# Myths, beliefs, and perceptions about COVID-19 in Ethiopia: A need to address information gaps and enable combating efforts

Yohannes Kebede[1]*, Zewdie Birhanu[1], Diriba Fufa[2], Yimenu Yitayih[3], Jemal Abafita[4], Ashenafi Belay[5], Abera Jote[6], Argaw Ambelu[7]

1 Department of Health, Behavior, and Society, Faculty of Public Health, Jimma University, Jimma, Ethiopia, 2 Department of Pediatrics and Child Health, Faculty of Medical Sciences, Jimma University, Jimma, Ethiopia, 3 Department of Psychiatry, Faculty of Medical Sciences, Jimma University, Jimma, Ethiopia, 4 Department of Economics, College of Business and Economics, Jimma University, Jimma, Ethiopia, 5 Department of English Language and Literature, College of Social Sciences and Humanities, Jimma University, Jimma, Ethiopia, 6 Faculty of Electrical and Computer Engineering, Jimma University, Jimma, Ethiopia, 7 Department of Environmental Health Sciences and Technology, Faculty of Public Health, Jimma University, Jimma, Ethiopia

* yohanneskbd@gmail.com

**Data Availability Statement:** All relevant data are within the manuscript and its Supporting Information files.

## Abstract

### Background

The endeavor to tackle the spread of COVID-19 effectively remains futile without the right grasp of perceptions and beliefs presiding in the community. Therefore, this study aimed to assess myths, beliefs, perceptions, and information gaps about COVID-19 in Ethiopia.

### Methods

An internet-based survey was conducted in Ethiopia from April 22 to May 04, 2020. The survey link was promoted through emails, social media, and the Jimma University website. Perceptions about COVID-19 have considered the World Health Organization (WHO) resources and local beliefs. The data were analyzed using Statistical Package for Social Science (SPSS) software version 20.0. Classifications and lists of factors for each thematic perception of facilitators, inhibitors, and information needs were generated. Explanatory factor analysis (EFA) was executed to assist categorizations. Standardized mean scores of the categories were compared using analysis of variance (ANOVA) and t-tests. A significant difference was claimed at p-value <0.05.

### Results

A total of 929 responses were gathered during the study period. The EFA generated two main categories of perceived facilitators of COVID-19 spread: behavioral non-adherence (55.9%) and lack of enablers (86.5%). Behavioral non-adherence was illustrated by fear of stigma (62.9%), not seeking care (59.3%), and hugging and shaking (44.8%). Perceived lack of enablers of precautionary measures includes staying home impossible due to economic challenges (92.4%), overcrowding (87.6%), and inaccessible face masks (81.6%)

**Funding:** The author(s) received no specific funding for this work.

**Competing interests:** The authors have declared that no competing interests exist.

and hand sanitizers (79.1%). Perceived inhibitors were categorized into three factors: two misperceived, myths (31.6%) and false assurances (32.9%), and one correctly identified; engagement in standard precautions (17.1%). Myths about protection from the virus involve perceived religiosity and effectiveness of selected food items, hot weather, traditional medicine, and alcohol drinking, ranging from 15.1% to 54.7%. False assurances include people's perception that they were living far away from areas where COVID-19 was rampant (36.9%), and no locally reported cases were present (29.5%). There were tremendous information needs reported about COVID-19 concerning protection methods (62.6%), illness behavior and treatment (59.5%), and quality information, including responses to key unanswered questions such as the origin of the virus (2.4%). Health workers were perceived as the most at-risk group (83.3%). The children, adolescents, youths were marked at low to moderate (45.1%-62.2%) risk of COVID-19. Regional, township, and access to communication showed significant variations in myths, false assurances, and information needs (p <0.05).

## Conclusions

Considering young population as being at low risk of COVID-19 would be challenging to the control efforts, and needs special attention. Risk communication and community engagement efforts should consider regional and township variations of myths and false assurances. It should also need to satisfy information needs, design local initiatives that enhance community ownership of the control of the virus, and thereby support engagement in standard precautionary measures. All forms of media should be properly used and regulated to disseminate credible information while filtering out myths and falsehoods.

## Introduction

The novel-coronavirus disease 2019 abbreviated as COVID-19 is currently a pandemic as declared by the World Health Organization (WHO) on January 30, 2020 [1]. The outbreak was first reported in late December 2019, when clusters of pneumonia cases of unknown etiology were found to be associated with epidemiologically linked exposure to the seafood market and untraced exposures in the city of Wuhan of China [2, 3].

The disease is highly infectious, and its main clinical symptoms include fever, dry cough, fatigue, myalgia, and dyspnea. Globally, 1 in 6 of the patients with COVID-19 develops to the severe stage, which is characterized by acute respiratory distress syndrome, septic shock, difficult-to-tackle metabolic acidosis, and bleeding and coagulation dysfunction [4, 5]. Epidemiologically, the distribution of the disease is exponentially growing across the globe. For example, on this date, June 9, 2020, the pandemics registered 7, 216,252 cases, and 409, 092 deaths in the world. Of the 3, 961,425 closed cases (10%) ended up in deaths. Ethiopia has become one of the COVID-19 affected countries as of March 12, the date on which one imported case was first detected. Since then, the infection by the virus has kept mounting. For example, on June 5, 2020, there were 2,152 total notified cases and 27 deaths in Ethiopia [1, 6, 7].

According to the WHO reports, the COVID-19 has no effective cure, yet early recognition of symptoms and timely seeking of supportive care and preventive practices enhance recovery from the illness and combat the spread of the virus. Older men with medical comorbidities are more likely to get infected, with worse outcomes [8–10]. Available evidence has shown that the

virus spreads from human-to-human mainly through respiratory droplets and body contacts. Contact with contaminated surfaces, hands, and touching of faces-eye-nose-mouth are predominant ways to get exposed to the infected droplets [11–14].

The battle against COVID-19 continues in Ethiopia. To guarantee the final success of stopping the virus, understanding myths and perceptions are so vital. Some questions require answers. For example, who do people think that they are most at risk? What are community perspectives about factors that facilitate the spread of the virus? What about perceived inhibitors? How scientifically accepted are these perceptions? Are the perceived facilitators or inhibitors correct or misperceived? Do people own the responsibility to fight the virus or externalize it? The answers to the above questions are of paramount importance to curb the pandemic by enhancing the probability of people's practicing the necessary precautions. Standard precautionary measures include avoidance of contact with surfaces, keeping physical distance, hand hygiene, respiratory hygiene, using sanitizers, and protective pieces of equipment [12–14].

The WHO recommends the risk communication and community engagement efforts to investigate and control "infodemics", myths, beliefs, and stigma so that the spread of the coronavirus would be effectively combated [10, 15, 16]. For example, the WHO reported risk perception, drinking alcohol, hot weather, and antibiotics related myths on COVID-19. Moreover, up-to-date information regarding causes, means of protection, modes of transmissions, diagnostic symptoms, and treatment/isolation procedures are relevant to withstand myths, beliefs, perceptions and support preventive efforts [12, 15, 17, 18].

The public health importance of COVID-19 has been recognized by the government of Ethiopia. There are movements to decentralize screening opportunities, quarantine, and treatment centers, and promoting precautionary measures. At the moment of the study, the government declared a state emergency in support of the precautionary measures, and has taken public measures such as the closure of schools, including universities; worked with public service outlets to install locally available preventive technologies, including handwashing machines; limiting the number of passengers in public transport, among others. Moreover, the ministry of health engaged in public awareness creation, risk communication, and community engagement tasks, and rallying voluntary activities. Now, addressing community beliefs, perceptions, and information gaps would reinforce the efforts to stop the virus. Therefore, this study aimed to assess community myths, beliefs, perceptions, and information needs via an online nationwide survey in Ethiopia.

## Methods and materials

### Study settings and designs

An internet-based online cross-sectional study was conducted in all regions of Ethiopia. At the time of the study, administratively, Ethiopia is divided into nine regional states and two federal cities. The regions have zonal divisions and district sub-divisions, with respective regional capitals and zonal/district towns. Internet services are rarely accessible at the district level. The Ethiopia 2020 population is estimated at 114,963,588 people in mid-year according to UN data. 21.3% of the population is urban (24,463,423 people in 2020). The median age in Ethiopia is 19.5 years [19]. The Ethiopian 2020 average literacy rate is 49.1% (lower among adults: male, 57.2; female, 41.1%, and higher among youths: male, 71.1%; female,67.8%) [20–21]. The survey tool was created through Google Form and the survey link was promoted through e-mail communications, social media (Facebook and LinkedIn), and the Jimma University website. The survey link was shared on April 22, 2020, and the responses were collected until May 04, 2020.

## Measurement and operationalization

The beliefs and perceptions tool about the spread and control of the virus were partly adapted from WHO resources [10, 15, 16, 22, 23]. Additionally, open-ended options were addressed to participants to report local beliefs and perceptions regarding COVID-19. Overall, four main themes of perceptions were asked: perceived facilitators for the spread of the virus (9 items), perceived inhibitors (9 items), information needs (7 items), risk labeling (8 items), access to communication resources (7 items), and socio-demographic variables, including residential regions and townships. Access to communication channels/platforms was measured by a score between 1 and 7 made on counting ownership or follow-up of television, official websites, social media, health workers, radio, friends/neighbors, and internet services/Wi-Fi. Townships referred to big towns/cities (regional/national capitals), zonal-level towns, and district-level (semi-urban/rural) towns. Perceived facilitators refer to people's perception of factors exacerbating the spread of COVID-19, while perceived inhibitors refer to people's misperceived or correctly perceived of factors that slow down the virus. These themes of perceptions were further categorized into a group of factors using explanatory factor analysis (EFA). A factor loading score of 0.4 was used as a cutoff value to retain items in each category [24]. Kaiser Mayer Olkin's (KMO>50%) indicated that the sample was adequate for executing EFA [24].

## Data analysis

Participants' online responses were first encoded on an Excel database and later exported to SPSS version 20.0 for analysis. Respondents' background variables and specific belief items are presented in the frequency tables. Standardized mean scores (0–100) and standard deviations were used to describe lists of categories of factors according to themes of perceptions they belonged to. One-way analysis of variance (ANOVA) and t-test were computed to compare the mean differences by region, township, and access to communication. A multi-response analysis was performed for every perception. A 95% confidence interval and a p-value of less than 0.05 were used to claim statistically significant association.

## Ethics approval and consent to participate

Jimma University, institutional review board approved the study. A reference number is IRB 00097/20.

## Results

### Socio-demographic characteristics of participants

A total of 929 participants from all regions of Ethiopia responded to this online survey questionnaire. Table 1 presents background information of the survey respondents. A majority of the respondents were in the age range of 30–39 years (50.8%), from Zonal towns (56.0%), and the Oromia region (56.6%).

### Perceived facilitating factors: How do people think about the spread of COVID-19?

**Classifications of facilitators.**   Table 2 presents categories and lists of perceived facilitators of COVID-19 spread with their respective prevalence. Explanatory factor analysis (EFA) produced two principal categories of perceived factors exacerbating COVID-19 in Ethiopia. The first category of factors was labeled as behavioral adherence, indicating that non-adherence to expected precautions is facilitating the virus; needing behavioral and social change.

**Table 1. Selected demographic characteristics of respondents, May 2020, Ethiopia.**

| Variables | Response category | Frequency | Percentage |
|---|---|---|---|
| Age in years | 18–29 | 285 | 30.7 |
| | 30–39 | 472 | 50.8 |
| | > = 40 | 172 | 18.5 |
| Gender | Male | 828 | 89.1 |
| | Female | 101 | 10.9 |
| Religion | Orthodox | 417 | 44.9 |
| | Protestant | 336 | 36.2 |
| | Muslim | 114 | 12.3 |
| | Others | 62 | 6.7 |
| Township | Big (regional capitals/national) city | 319 | 34.3 |
| | Zonal level town | 520 | 56.0 |
| | District level/Semi-urban/town | 90 | 9.7 |
| Region | Oromia | 526 | 56.6 |
| | Addis Ababa | 139 | 15.0 |
| | SNNP* | 103 | 11.1 |
| | Amhara | 52 | 5.6 |
| | Tigrai | 49 | 5.3 |
| | Other regions | 60 | 6.5 |

*SNNP: Southern Nations and Nationalities People

The items that contributed to behavioral non-adherence include that people still shake each other's hands, do not seek care for symptoms suggestive of COVID-19, use crowded transport means, are not being screened for flu-like symptoms, and fear of stigma with respective decreasing order of factor loading scores (0.714–0.503). The second category of perceived facilitating factors was the lack of enabling environmental conditions that are supposed to support adaptations of precautionary measures. The lack of enablers was made up of economic reasons that challenge stay at home principle, overcrowded living/working conditions, absence of PPE like face masks, and sanitizers with decreasing order of factor loading scores (0.786–0.718). The behavioral non-adherence and lack of enablers related factors explained an overall variance of perceived facilitators of the virus by 48.8%.

**Table 2. Perceived categories and lists of exacerbating factors of COVID-19, May 2020, Ethiopia.**

| Perceived COVID-19 exacerbating factors | Principal components and factor loading score | | Descriptive statistics | |
|---|---|---|---|---|
| | Behavioral non- adherence | Lack of enabling environment | Freq. | % (95% CI) |
| People fear stigma and bias related to COVID-19 | .503 | | 584 | 62.9 (59.7,65.9) |
| People still use crowded transportation means | .654 | | 562 | 60.5 (57.4,63.3) |
| People with flu-like symptoms are not well screened for COVID-19 | .638 | | 551 | 59.3 (56.1, 62.5) |
| People do not often seek care for symptoms that looks like COVID-19 | .681 | | 481 | 51.8 (48.7, 55.1) |
| People still hug and shake each other's hands while greeting | .714 | | 416 | 44.8 (41.5. 47.8) |
| People do not stay at home for economic and social reasons | | .786 | 858 | 92.4 (90.6,94.2) |
| People still live and work in a very crowded condition | | .705 | 814 | 87.6 (85.4, 89.6) |
| People do not have PPE like face masks | | .727 | 758 | 81.6 (78.9, 84.0) |
| People do not have hand rub alcohol or sanitizers | | .718 | 735 | 79.1 (76.3,81.6) |

Notes: KMO = 81.9%); Variance Explained (VE = 48.8%); and PPE: Personal Protective Equipment.

**Prevalence of facilitators.** Descriptive statistics columns in Table 2 indicate the prevalence of specific factors in categories of facilitators they belonged to. Accordingly, the prevalence of specific factors that contributed to behavioral non-adherence ranged between 584 (62.9%) and 416 (44.8%). The fear of stigma and people's continued use of suffocated transport means accounted for a higher extent of non-adherences. The magnitudes of lack of enablers that would support behavioral adherence range between 858 (92.4%) and 735 (79.1%). Staying at home is impossible for economic reasons (92.4%) and living/working in overcrowded conditions accounted for a major share of the lack of enablers. Deterring environmental conditions were perceived at a higher prevalence than behavioral non-adherence, indicating a high tendency of externalizing factors that could aggravate a spread of COVID-19 in the community. There were 53 (5.7%, 95%CI:4.3%-7.4%) factors reported by respondents as unknown.

## Perceived inhibiting factors: How do people think about the slow down of COVID-19?

**Classifications of inhibitors.** Table 3 presents categories and lists of perceived inhibitors of COVID-19 spread with their respective prevalence. EFA produced three principal categories of perceived factors inhibiting COVID-19. Two of the three categories were misperceived (myths and false assurances), while one was correctly perceived inhibitor. The myths category was composed of factors that are believed to inhibit the virus without having been scientifically proven. In this case, the myths include: eating selected foods (garlic, onion, ginger, etc) for prevention and cure; perceived religiosity (perceiving oneself as an effective religious man/woman in controlling challenges); drinking alcohol; people's perceived confidence that they owned effective traditional medicines that were, however, not clinically confirmed; and living in hot weather. The factor loading scores in respective order ranged between 0.764–0.488. The second category of perceived inhibitors was still local sayings that were often related to false

**Table 3. Perceived categories and lists of inhibiting factors of COVID-19, May 2020, Ethiopia.**

| Perceived COVID-19 inhibiting factors | Principal components and factor loading scores | | | Descriptive statistics | |
|---|---|---|---|---|---|
| | Myths | Invulnerability (false assurances) | Engaged in precautions | Freq. | % (95% CI) |
| We are religious enough to control COVID-19 | .496 | | | 508 | 54.7 (51.5, 58.0) |
| We are eating garlic, onion, honey among others to prevent COVID-19 | .764 | | | 455 | 49.0 (45.7, 54.3) |
| The weather we live-in is too hot for coronavirus to survive | .488 | | | 242 | 26.0 (23.6, 29.1) |
| We are eating garlic, onion, honey among others to cure COVI-19 | .728 | | | 227 | 24.4 (21.6, 27.2) |
| We believe we have traditional medicine against COVID-19 | .511 | | | 165 | 17.8 (15.5, 20.3) |
| We are drinking alcohol to protect against COVID-19 | .676 | | | 140 | 15.1 (12.9, 17.3) |
| There are no locally reported COVID-19 cases so far | | .770 | | 343 | 36.9 (33.8, 39.7) |
| We live far away from COVID-19's rampant areas | | .661 | | 274 | 29.5 (26.8, 32.4) |
| Engaged in standard precautions measures of COVID-19 | | | .775 | 159 | 17.1 (14.9, 19.7) |

Notes: KMO = 77.3%, Variance explained (VE = 54.6%)

assurances that people were protected from COVID-19 (unlike myths, the second category of beliefs may not need scientific approval or disapproval). The category consisted of two main beliefs: "*we live far away from COVID-19 rampant areas*" and "*there are no locally reported COVID-19 cases so far*", with factor loading scores (0.770–0.661). The beliefs looked false assurances in that people perceive themselves as living out of a risk zone that is an impression of invulnerability. The third, correct, and promotable category of perceived inhibitors was a single item about people having been engaged in standard precautions (factor score loading = 0.775). Factors related to the above three categories explained an overall variance of perceived inhibitors by 54.6%, indicating the presence of several other unreported myths and unhealthy beliefs that need further assessment.

**Prevalence of inhibitors.** Descriptive statistics columns in Table 3 indicate the prevalence of perceived inhibitors. Myths and false assurances were the most prevalent perceived inhibitors of the spread of COVID-19 compared to the perception that engagement in precautionary measures protect from exposure to and spread of the virus. Specifically, perceived religiosity, effectiveness of selected foods, and perceived protectiveness of hot weather were the commonest myths, accounting for 508 (54.7%), 455 (49.0%), and 242 (26.0%), respectively. Beliefs that there were no locally reported cases of COVID-19, and the specific localities where respondents are currently living are far away from coronavirus rampant areas contributed to 343 (36.9%) and 274 (29.5%) respective prevalence of false assurances. On the other hand, the prevalence of a perception that the spread of COVID-19 would be controlled as a result of people's active engagement in standard precautionary measures was as low as 159 (17.1%). Overall, false beliefs and myths were more rampant than accurate perceptions about factors that potentially inhibit the spread of COVID-19. About 153 (16.5%, 95%CI:14.2–18.8%) respondents reported that they were unsure of other factors which potentially inhibit the distribution of COVID-19 given the virus is newly introduced

## Perceived information needs: What do people want to learn more about COVID-19?

**Classifications of information needs.** Table 4 presents the information needs of the community concerning COVID-19. The EFA generated four categories of information needs. The first category of information needs was related to prevention that is composed of how to surely protect from the virus, exhaustive transmission modes, and distinguishable symptoms. The factor loading score ranged from 0.816–0.842. The second category was related to illness and treatment. Specifically, in this category, people want to know about the nature of the treatment, details about isolation and quarantine, what to do when at risk or as a high-risk group, and procedures to follow when symptomatic (factor loading range, 0.534–0.786). The third category was related to quality, including true and up-to-date, and change provoking information. The fourth was diverse information needs, ranging from the need to know about the readiness of the health facility to confirmation of the origin of the virus.

**Magnitude of information needs.** Descriptive statistics columns in Table 4 indicate the prevalence of the information needs according to their respective categories. For example, the magnitude of people who need to prevent the virus by knowing mechanisms of protection, exhaustive transmission modes, and diagnostic symptoms were 605 (65.2%), 554 (59.6%), and 529 (56.9%), respectively. The highest information needs about COVID-19 was related to illness behavior and treatment, for example, isolation and quarantine accounted for 611 (65.8%). In terms of quality information, about 27 (2.9%) of people needed to know about how to alleviate community reluctance. There were mixed communication needs, 14 (1.5%).

Table 4. Perceived categories and lists of information needs about COVID-19, May 2020, Ethiopia.

| Perceived information need factors about COVID-19 | Principal components and factor loadings scores | | | | Descriptive statistics | |
|---|---|---|---|---|---|---|
| | Preventive | Illness and treatment | Quality information | Diverse questions | Freq. | % (95% CI) |
| How to protect from COVID-19 | .816 | | | | 605 | 65.2 (62.2, 68.2) |
| Exhaustive transmission modes | .839 | | | | 554 | 59.6 (56.3, 62.9) |
| Distinguishable symptoms | .842 | | | | 529 | 56.9 (53.9, 60.3) |
| Details on isolation and quarantine | | .683 | | | 611 | 65.8 (62.8, 68.9) |
| What to do when they or someone become symptomatic (illness behavior) | | .534 | | | 581 | 62.5 (59.3, 65.7) |
| Nature and process of treatment | | .786 | | | 552 | 59.4 (56.4, 62.4) |
| What to do with risk factors or as a risk group | | .587 | | | 412 | 44.3 (41.1, 47.6) |
| Change provoking information** | | | .643 | | 27 | 2.9 (1.8, 4.1) |
| True and update information | | | .867 | | 12 | 1.3 (0.5, 2.0) |
| Diverse information needs* | | | | .907 | 14 | 1.5 (0.6, 2.2) |

Notes: Kaiser Mayer Olkin's measure of sampling adequacy (KMO = 80.5%), Variance explained (VE = 65.4%).

* Diverse information need: learn about capacity and readiness of the health facilities to manage in transmission peaks, costs related to treatment services, community screening service, want to differentiate the origin of the disease itself as to whether it is a Wrath of the Creator or biological weapon, need praying, among others.

** Change provoking information: bridging knowledge to behavior change, Alleviation of reluctance to precautions, messages involving a specific audience, increasing vulnerability perception, repeatedly accessing with messages, enforcement of laws that save guard lives, implementations of command posts in favor of combating COVID-19, how the jobless can be economically supported, where to get sanitizers, among others.

## Perceived risk labeling: Who is perceived to be more vulnerable?

Table 5 presents COVID-19 risk labels and groups. The study showed that 656 (70.8%, 95% CI: 68.0%, 73.1%) of the community felt COVID-19 as a dangerous disease. The perception of vulnerability to an infection of COVID-19 looked somewhat lower, 536 (57.8%, 95% CI: 54.6%, 61.1%). The community perceived that health workers (83.2%), people with underlying illnesses (78.8%), and elderly people (76.3%) are at high-risk of COVID-19. Age ranges between 0–30 years old were classified into low-moderate risk (45.1–62.2%).

Table 5. Perceived COVID-19 risk groups and labels, May 2020, Ethiopia.

| Perceived high-risk groups | Descriptive statistics | |
|---|---|---|
| | Freq. | % (95% CI) |
| Health workers | 773 | 83.2 (80.7, 85.7) |
| People with underlying illness conditions | 732 | 78.8 (76.1, 81.4) |
| Elderly people | 709 | 76.3 (73.6, 78.9) |
| Adults (30–50 years old) | 597 | 64.3 (60.9, 67.3) |
| Youth (16–29 years old) | 578 | 62.2(59.1, 65.2) |
| Pregnant women | 552 | 59.4 (56.5. 62.5) |
| Adolescents (10–15 years old) | 448 | 48.2 (45.0,51.3) |
| Children (0–9 years old) | 419 | 45.1 (41.9,48.3) |

**Table 6. Descriptive statistics and regional ranges for perceptions and needs, May 2020, Ethiopia.**

| Beliefs and information need categories | Median | %mean(±SD) | Regional ranges | p-value |
|---|---|---|---|---|
| **Perceived facilitators (overall)** | 66.7 | 69.5 (±15.6) | 62.8–73.5 | 0.239 |
| Behavioral non-adherence | 60.0 | 55.9 (±11.2) | 49.0–61.0 | 0.323 |
| Lack of enabling conditions | 85.5 | 86.5 (±6.5) | 80.1–89.2 | 0.262 |
| **Perceived inhibitors (overall)**\*\* | - | - | - | - |
| Misperceived inhibitor: Myths | 33.3 | 31.6 (±11.2) | 24.8–36.9 | 0.002\* |
| Misperceived inhibitor: False assurance | 36.3 | 32.9 (±4.6) | 25.5–49.5 | <0.001\* |
| Engagement in standard precautions | 17.0 | 17.1 (±2.5) | 6.7–22.5 | 0.146 |
| **Information need (overall)**\*\*\* | 58.3 | 59.3 (±3.4) | 52.4–65.3 | 0.031\* |
| Prevention related | 66.7 | 62.6 (±8.1) | 50.6–66.7 | 0.021\* |
| Illness and treatment-related | 53.2 | 59.5 (±8.9) | 53.1–63.6 | 0.317 |
| Quality information | 3.6 | 2.4 (±1.4) | 0.0–2.4 | 0.590 |
| Mixed information need | 1.5 | 1.7 (±0.8) | 1.1–4.1 | 0.443 |

\* Statistically significant at p <0.05 (two-tailed)

\*\*Overall perceived inhibitor has two misperceived (myths and false assurances) and one correctly perceived (engaged in standard precautions) aspect, needing no further merging for an overall score.

\*\*\*The overall mean of information needs to exclude the two dimensions-quality and mixed needs because of extreme values.

## Description of overall perceptions of facilitators, inhibitors, and information needs

The above mentioned specific beliefs about inhibitors, facilitators, and information needs were merged based on categories the items belonged to (as referred to in Tables 2–4). Standardized means scores ranging from 0–100 were calculated for all categories of perceptions and information needs. Table 6 provides the details of overall standardized mean scores and regional ranges. Without noting significant variations in regions, there was high (59.5%) perceived nationwide information needs about illness behavior and treatment procedures (p = 0.317). Likewise, the lack of enablers and behavioral non-adherence that were perceived as facilitators of the spread of the virus were high, 86.5% (p = 0.262) and 55.9% (p = 0.323), respectively.

## Spatial distributions of the perceptions: variations by regions and townships

**Regional distribution and variation.** One-way ANOVA showed significant regional differences, particularly on factors perceived to inhibit the spread of the virus and information needs. Specifically, the variations were on myths (F = 3.75, p = 0.002), false assurances (F = 6.57, p <0.001), and overall (F = 2.48, p = 0.031) and preventive information needs (F = 2.68, p = 0.021). Moreover, Fig 1 shows a specific regional concentration of the perceptions about the spread and control of the virus. Accordingly, a slight but significant higher prevalence of myths was observed in Addis Ababa compared to Tigrai and Oromia regions, with MD(95%CI) of 13.4 (1.0,24.9%) and 9.1 (1.3,16.9%). There were higher scores false assurances (an impression of invulnerability) in the Southern region compared to Oromia and Addis Ababa. The variation ranged 18.7% (7.4–30.0%, p <0.001) and 24.0% (10.3%-37.6%, p<0.001) with Oromia and Addis Ababa, respectively. Information needs were highest in Southern and Oromia regions compared to Addis Ababa, with respective MD = 16.4, p = 0.038, and 12.4, p = 0.028.

**Township distribution and variation.** Respondents' township showed significant differences in myths (F = 10.62, p <0.001), overall information need (F = 6.91, p = 0.001), and

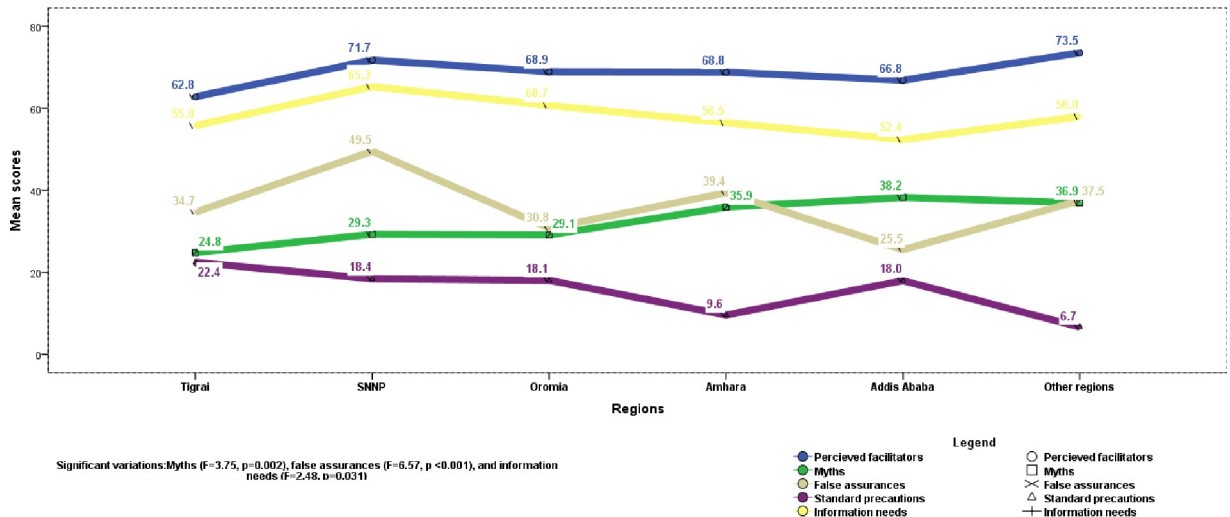

**Fig 1. Diagram of regional distribution of perceptions about COVID-19, May 2020, Ethiopia.**

particularly preventive information (F = 5.23, p = 0.006), Fig 2 shows diagrammatic township distribution of the perceptions concerning the virus. Hence, myths that are perceived to inhibit the spread of the virus were more prevalent in big cities/towns including Addis Ababa compared to the zonal (MD (95CI%) = 8.8% (4.0–13.6%), p <0.001), and district/semi-urban (MD = 9.4% (1.4–17.4%, p = 0.015) towns. Community residing in the zonal and district towns felt that there was higher information need in their community, particularly about protection ways compared to big towns/cities, with respective MD of 8.3% (1.2–15.4%, p = 0.015) and 16.4% (4.5–28.3%, p = 0.003).

**Communication resources and perceptions.** Fig 3. presents variations of COVID-19 related perceptions by the number of communication sources accessed. One-way ANOVA revealed significant differences in mean scores of perceived facilitators, inhibitors, and information needs about a spread and control of COVID-19 by the number of a mix of

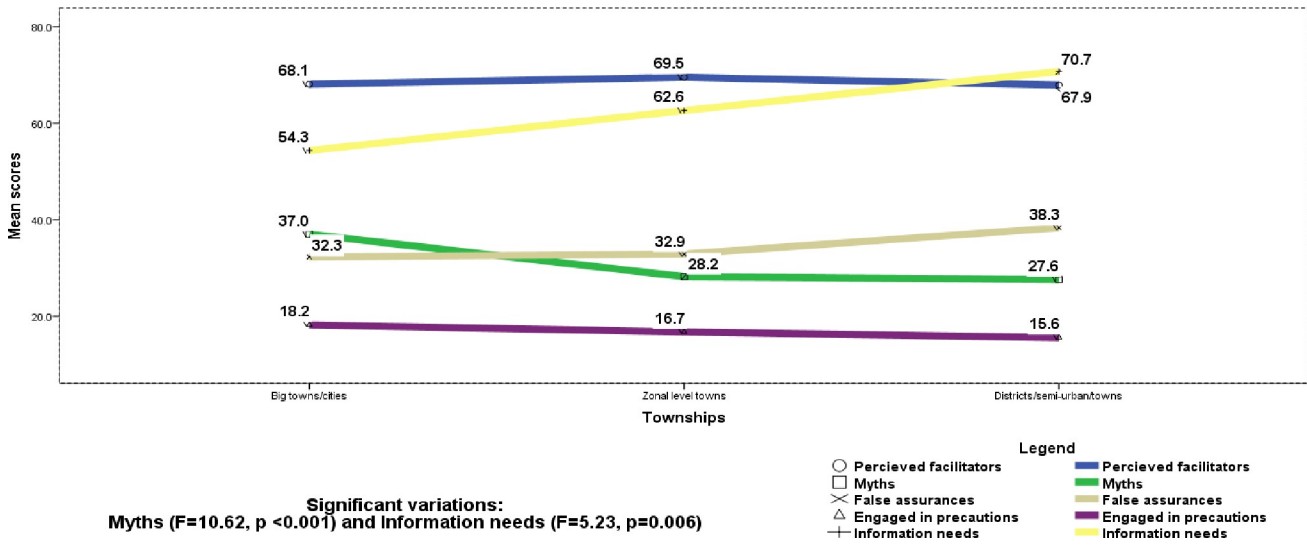

**Fig 2. Diagram of township distribution of perceptions about COVID-19, May 2020, Ethiopia.**

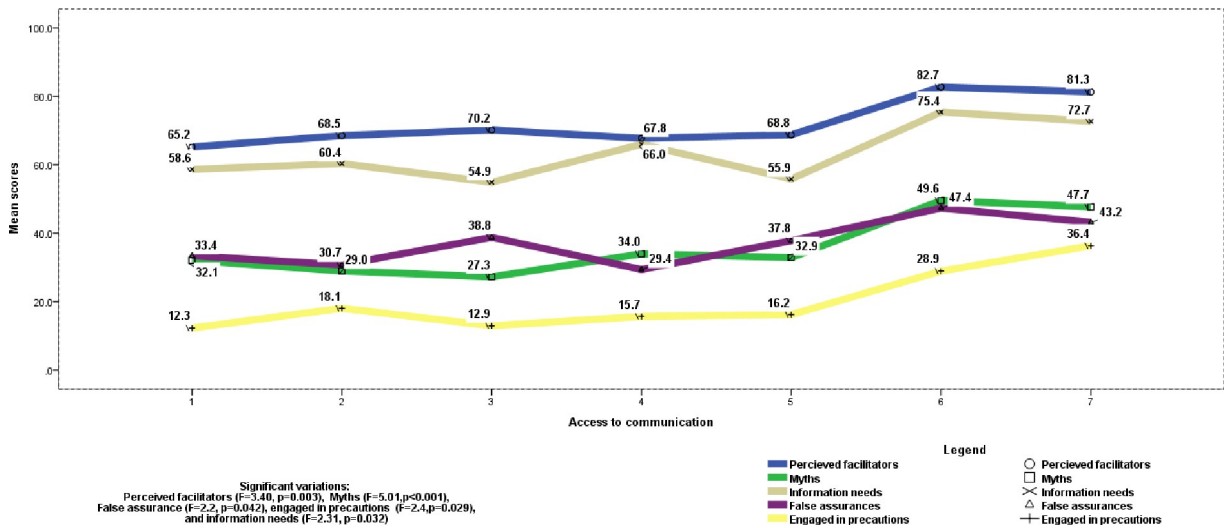

**Fig 3. Diagram of distribution of perceptions by access to communication platforms, May 2020, Ethiopia.**

communication channels accessed. Specifically, Overall perceived facilitators (F = 3.40, p = 0.03), behavioral non-adherence (F = 3.47, p = 0.002), myths (F = 5.01, p <0.001), false assurances (F = 2.2, p = 0.042), engagement in precautions (F = 2.40, P = 0.029), and overall information need (F = 2.31, p = 0.032). Access to television, official websites, social media, health workers, radio, friends/neighbors, and internet/Wi-Fi platforms/channels scored. Accordingly, for most of the variables with significant differences by communication sources, access to only 1 or2 sources led to lower means of perceptions compared to access to 6 sources. This indicated that the number of communication channels accessed may not be as important as the quality of messages they carried in affecting information needs and beliefs.

**Perception of threat and perceived facilitators, inhibitors, and information needs.**
Community perception of threat (the result effect of perception of susceptibility to a dangerous virus) from COVID-19 showed a statistically significant mean difference (MD) in scores of perceived lack of enabling environment that facilitate a spread of the virus (MD (95% CI) = 3.43 (0.11,6.77), p = 0.043), presence of myths (MD (95%CI) = 4.75(1.13,8.36), p = 0.010), perceived engagement in standard precautions (MD (95% CI) = 12.15(7.35,16.94), p <0.001), overall information needs (MD (95%CI = 4.47(1.67,7.27), p = 0.002), preventive information need (MD (95%CI) = 6.26(2.75,11.65), p = 0.023), and treatment procedures related information needs (MD (95%CI) = 7.10(2.77,211.41), p = 0.001).

## Discussion

This online survey has generated pertinent findings of nationwide community perceptions concerning factors that facilitate and inhibit a spread of COVID-19, risk labeling, and information needs in Ethiopia. The perceived factors were aligned into the following main categories: behavioral adherence, lack of enabling environmental conditions, myths, false assurances, engagement in standard precautions, and information needs about prevention, illness behavior and treatment, including answers to diverse questions related to the origin, a spread and control of the coronavirus. Each perceived factor was discussed step by step as follows:

This study found a moderate perception of severity by the community, 70.8%, while, somewhat low perceived vulnerability, 57.8%. This indicates the community's perception of risk should be increased further. The perceptions were measured by a single item for each. There

were two forms of risk labeling and groups in the community: As perceived by the community, young people below 30 were perceived as a low-moderate risk with an increasing order: 0–9 years old (45.1%), 10–15 years old (48.2%), 16–29 years old (62.2%), and 30–50 years old (64.3%). Health workers, people with underlying illnesses, and the elderly were perceived as high-risk groups with the respective prevalence of 83.2%, 78.8%, and 76.3%. Perhaps, the high-risk groups perceived by the community, in this study, were consistent with that of the WHO. According to WHO, frontline health workers, people with underlying illness, and elderly people are high-risk groups [22, 23]. The correct perception of the high-risk group is important for giving protection priorities against infection by COVID-19. However, this study reported that children, adolescents and youths were relatively perceived as lower risk groups (45.1%, 48.2%, and 62.2%, respectively). This would be concerning to the control efforts to some extent. We argue that those who were perceived as being at low-risk would act as reservoirs for a spread of COVID-19 for a couple of reasons: one, about 63% of the Ethiopian population aged < 25, with a median age of 19.5, and these segments pass time searching for jobs like daily labors [20, 21]. Two, in one of the previous studies conducted in Ethiopia, 179 (72.5%) of respondents knew that the elderly and people with underlying illnesses are high-risk groups, while only 15 (6.1%) knew that young adult people must engage in precautions just like any other segment [25] Therefore, some enforcement needs to control a potential contribution of youths in the transmission loop as the current perception of risk groups stands.

Factors that were perceived to exacerbate the spread of the virus were teamed up into two thematic categories: behavioral non-adherence (55.9%), and lack of enabling environmental conditions (86.5%). Behavioral non-adherence, in this case, referred to individuals and social ignorance, disregard, and lack of commitment to convert standard precautionary measures that seem to be under the control without needing much material support. The ignorance and lack of commitment were illustrated by the following community's experiences: people still hug each other and shake hands while greeting, do not often seek care while showing symptoms that look like COVID-19, still feel comfortable to use crowded unventilated transport means, and fear stigma-related to the virus. Interestingly, the use of crowded/unventilated transport means was not only due to lack, but rather it also was involved in behavioral non-adherence. Theoretically, people often rationalize their engagement in preventive actions, and rationalities should be carefully studied and justified [26]. On the other hand, lack of enabling environments is about condition and resource factors whose presence or absence enable people to take precautionary actions. Some of them can be illustrated as such people cannot stay at home for economic and social reasons, do not have personal protective equipment (PPE) like face masks, do not have hand-rub alcohol or sanitizers, and still live and work in crowded condition. In this study, the magnitudes of both behavioral non-adherence and perceived lack of environmental conditions were high, irrespective of regions and townships. Behavioral and communication theories indicate that people's perceived lack of resources negatively affects actual practices [26]. Nonetheless, the high prevalence of perceived facilitators signals two main urgencies. One, it suggests strong work to alleviate behavioral non-adherence, and lack of enablers that facilitate the spread of the virus. Two, even a higher perceived lack of enabling conditions looks concerning given that it may lead people to externalize the capacity to control the virus, while ignoring to their personal efforts. Thus, to convert this perception into opportunity, local initiatives that support engagement in standard precautions should address the locally perceived barriers, and enhance a shared responsibility and community ownership to involve in efforts of combating COVID-19 [10].

Factors that were perceived as inhibitors of the spread of the virus were classified into three: false assurances (32.9%), myths (31.6%), and engagement in standard precautions (17.1%). Interestingly, the first two of the three factors were wrongly perceived inhibitors, that was why

we labeled them myths and false assurances. False assurances were impressions of invulnerabilities, and characterized by people's perception that they were living out of the COVID-19 risk zone. In the current study, the two main false assurances were the perceived absence of locally reported COVID-19 cases and residence far away from COVID-19 rampant areas. One study from the Kingdom of Saudi Arabia presented walking through sanitized gates could give a false sense of protection and potentially deceit the passersby from taking the recommended preventive actions [27]. In the current study, myths include: perceived effectiveness of religiosity (54.6%), food items (49.0%), living in hot weather (26.0%), traditional medicines (17.8%), and drinking alcohol (15.1%) to protect from COVID-19. WHO myth busters list out most of the misperceptions presented in this study, indicating that these were globally shared altogether with the pandemic [15]. Pieces of evidence indicate that myths or misperceptions like denial of the presence, and misperceived causes, transmissions modes, and protection ways can set back preventive and control efforts in times of the pandemics of HIV, Zika virus, Yellow fever, and Ebola, unless traced and addressed [28–31]. The magnitude of the correctly perceived factor (engagement in standard precautionary measures) for inhibiting the spread of COVID-19 was too low (17.1%), demanding hard work to promote this perception until a larger segment of the community embraces an accurate reason for protection from the virus.

The finding from the current study revealed that the majority of the information needs were related to protection methods that are symptoms, mode of transmission and prevention (56.9%-65.2%), and procedures to be followed when someone feels ill from COVID-19 or at risk of contracting it, including isolation, quarantine, and treatment (44.3–65.8%). Particularly, people want to access information about isolation and quarantine–how it works (65.8%), and what to do when someone becomes symptomatic (65.2%). One study in 2018 on health information needs during the outbreak of Ebola showed that there was a need to an uninterrupted access to an up-to-date information including about causes, transmission modes, cures, the readiness of health facilities, and even the role of government [31]. Some studies related to illness behavior and drug repurposing from Pakistan and Saudi Arabia revealed that misinterpretation or misinformation (less quality or inadequate) about treatment/medicines that were delivered by press, electronic and social media has been leading to self-medication by chloroquine, hydroxychloroquine, and Ivermectin as COVID-19 cure [32, 33]. Interestingly, though minor proportion, there were people who sought quality and change provoking information that is true, up-to-date, how it is possible to alleviate ignorances that exist in the community regarding the adaptation of precautions of COVID-19, at the presence of basic knowledge. Cognitive dissonance theory recommends audience-specific messages that satisfy the information needs to close the gaps between knowledge and practices [34]. This study found out that some questions were left unanswered about COVID-19, one of these was the need for information about the origin of the virus. No matter the reported magnitude of such a question, providing convincing responses would enhance the uptake and support for preventive and treatment efforts. For example, one study from Pakistan reported that some recognized political figures claimed conspiracy (the virus was aimed to affect Muslim countries) as to the origin of the virus and raised public hesitancy to the COVID-19 vaccine which is under development [35].

In this study, significant regional differences were observed on myths, false assurances, and preventive information needs. Specifically, a slightly higher magnitude of myths and lower information need was observed in Addis Ababa. From the date of onset until 9 June 2020, Addis Ababa constituted about 3/4th (1,625 of 2,156 cases) of an accumulation of people with COVID-19, as referred to in most of the daily notification note on COVID-19 situational updates [36, 37]. Addis Ababa is located at the center of Ethiopia, geographically, politically, and economically. Thereof, it has an enormous connection with most Ethiopian regions and

towns, which would later lead to a massive spread of the virus to the rest of the regions, due to myths. Additionally, this study found variations in the distribution of myths based on the township, a significantly higher accumulation was observed in big towns than zonal or district towns. Therefore, serious attention needs to be paid to further understand and clear the myths, particularly in Addis Ababa and other big cities/towns in Ethiopia. False assurances that are perceived to inhibit the spread of the virus were common in the Southern region compared to others. Crudely speaking, the false assurances related to the perception of living out of risk zones may seem to go with the prevalence of COVID-19 cases reported in the Southern region. covid-19 case distributions notified by the ministry of health currently indicated, only 15 of 2,156 (0.70%) of cases and zero death were found in the southern region until June 9, 2020 [37]. However, there is no warranty that the virus has not yet been spread across the region, given the testing centers or testing capacity have not yet reached out well in Ethiopia at the moment of the study. The perceptions that there were no locally reported COVID-19 cases and people were living far away from case rampant areas may remain deceitful. Concerning information gaps, southern regions, and zonal and district towns showed higher needs, particularly for preventive information. Currently, a vaccine is one of the most common topics people want to get informed about, but largely affected by conspiracy theories as one of the studies from Pakistan revealed [35].

The above records about perceptions justify that the community's readiness and responses against a spread of the virus would not withstand the fast-growing rate of infection, suggesting a lot of risk communication and community engagement works. There were a couple of reasons to support this idea. First, the magnitude of the correctly perceived inhibitor (engagement in precautions) of the spread of the virus was as low as 17.1%. Second, there were high perceived magnitudes of behavioral non-adherence and lack of required resources regarding efforts to combat COVID-19. Third, myths and false assurances were rampant.

## Limitations of the study

This online questionnaire survey gathered nationwide data capturing community perceptions and experiences that are helpful to have input for risk communication and community engagement. In times of crisis like this pandemic, an online survey looks partly cost-effective and partly ethical. Nonetheless, the study was not without limitations. For example, as with any other online survey, the respondents were relatively educated ones who had access to internet services. On top of this, the perceptions were analyzed from participants' responses about what people in their locality think, feel, and need about a spread and control of COVID-19. This is an entirely proxy indicator for community perceptions and information needs. Although the study was nationwide, participation from some regions was limited compared to others. Perhaps, extended data collection period would have increased their involvement and representations. Moreover, the current study did not report correlations of the perceptions and community practices. The findings were not well compared with literature due to the absence of similar studies. Nonetheless, we assert that the findings are pertinent to address information gaps and support preventive and treatment efforts. To the best of our knowledge, this study is the first kind of community perceptions and myths on COVID-19 in Ethiopia.

## Conclusions

This assessment of the community's perceived factors facilitating and inhibiting a spread of COVID-19, risk group labeling, and information needs provides important signals to control the spread of the virus. There were substantial magnitudes of perceived behavioral non-adherence, lack enabling resources, myths, false surety, information needs, and low perceived

adaptations of standard precautions. These sum up to a high likelihood of ignorance of protective measures and externalization of the capacity to control the virus, thereby facilitating the spread of the virus. A lot of myths and false assurances were perceived that were wrongly labeled as inhibitors of the spread of the virus such as perceived religiosity, perceived effectiveness of selected food/spice items, living in hot weather environment, traditional medicines, drinking alcohol, and residence out of risk zone. Regional and township variations in magnitudes of myths, false assurances, and information gaps suggest a need for disproportionate and local framing of communication and interventions that enhance community ownership of the fight against the pandemic. Myths and false assurances should urgently be addressed in higher and lower COVID-19 incidence settings, respectively. Zonal and district towns had higher information needs. Access to multiple mixes of communication channels that deliver quality messages is required to fill information needs rather than mere number of sources. People's commonest information needs include: how surely people can protect from the virus, isolation and quarantine, and procedures that a symptomatic person needs to follow to keep onself healthier. Though health workers, elderly people, and people with underlying illnesses were perceived high-risk groups as labeled by WHO, perceiving adolescents and youths as low-moderate risk groups would be challenging in a country with a high percentage of young population, like Ethiopia. The young population deemed special attention so that they would actively participate in the prevention efforts. The risk communication and community engagement efforts should: 1) consider regional and township variations in myths and false assurances, 2) investigate more beliefs that could facilitate/inhibit the spread of the virus, 3) satisfy the information needs, 4) design local initiatives that enhance community ownership of tasks of controlling the virus, and thereby support and advocate engagement in standard precautionary measures, and 5) properly utilize media in filtering and disseminating credible information amid increasing volume of disparate falsehoods against COVID-19, supported with the appropriate regulatory system.

## Supporting information

**S1 Questionnaire.**
(DOCX)

## Acknowledgments

We express our heartfelt thanks to all individuals who participated in the study: respondents, individuals who have supported data collection across the regions, and professionals who assisted the operations of this online survey.

## Author Contributions

**Conceptualization:** Yohannes Kebede, Zewdie Birhanu, Diriba Fufa, Yimenu Yitayih, Argaw Ambelu.

**Data curation:** Yohannes Kebede, Zewdie Birhanu, Diriba Fufa, Yimenu Yitayih, Argaw Ambelu.

**Formal analysis:** Yohannes Kebede, Zewdie Birhanu, Argaw Ambelu.

**Investigation:** Yohannes Kebede, Zewdie Birhanu, Diriba Fufa, Yimenu Yitayih, Abera Jote, Argaw Ambelu.

**Methodology:** Yohannes Kebede, Zewdie Birhanu, Diriba Fufa, Yimenu Yitayih, Argaw Ambelu.

**Project administration:** Yohannes Kebede, Zewdie Birhanu, Diriba Fufa, Jemal Abafita, Ashenafi Belay, Abera Jote, Argaw Ambelu.

**Resources:** Yohannes Kebede, Zewdie Birhanu, Diriba Fufa, Yimenu Yitayih, Jemal Abafita, Ashenafi Belay, Abera Jote, Argaw Ambelu.

**Software:** Yohannes Kebede, Zewdie Birhanu, Diriba Fufa, Jemal Abafita, Ashenafi Belay, Abera Jote, Argaw Ambelu.

**Supervision:** Yohannes Kebede, Zewdie Birhanu, Diriba Fufa, Jemal Abafita, Abera Jote, Argaw Ambelu.

**Validation:** Yohannes Kebede, Zewdie Birhanu, Diriba Fufa, Jemal Abafita, Abera Jote, Argaw Ambelu.

**Visualization:** Yohannes Kebede, Zewdie Birhanu, Diriba Fufa, Yimenu Yitayih, Jemal Abafita, Ashenafi Belay, Abera Jote, Argaw Ambelu.

**Writing – original draft:** Yohannes Kebede.

**Writing – review & editing:** Yohannes Kebede, Zewdie Birhanu, Diriba Fufa, Yimenu Yitayih, Jemal Abafita, Ashenafi Belay, Abera Jote, Argaw Ambelu.

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
