## [Decision Letter · Decision Letter 0]

29 Oct 2020

PONE-D-20-17606

Myths, beliefs, and perceptions about COVID-19 in Ethiopia:  The need to address information gaps

PLOS ONE

Dear Dr. Kebede,

Thank you for submitting your manuscript to PLOS ONE. After careful consideration, we feel that it has merit but does not fully meet PLOS ONE’s publication criteria as it currently stands. Therefore, we invite you to submit a revised version of the manuscript that addresses the points raised during the review process.

Thank you for submission to Plos One. This manuscript has been assessed by two experts and found it interesting. However, authors raised serious concerns in the readability, methodology and results. I invite you to consider these suggestions in point-by-point manner. Since COVID-19 is associated with massive infodemic resulting in various false beliefs, this manuscript is timely and well needed during this time, particularly in African regions. Myths and misleading beliefs during the pandemic has also raised serious concerns such as vaccine hesitancy, self-medication, inappropriate use of devices, drug shortages and price hikes. Various drafts have raised this issue but scientific evidence is currently lacking. I will suggest authors to consider following recently published articles in introduction and discussion section as most of the author`s claim are supported by them. 1. Threat of COVID-19 Vaccine Hesitancy in Pakistan: The Need for Measures to Neutralize Misleading Narratives (ajtmh.org/content/journals/10.4269/ajtmh.20-0654), 2. Misinformation in wake of the COVID-19 outbreak: Fueling shortage and misuse of lifesaving drugs in Pakistan (https://www.cambridge.org/core/journals/disaster-medicine-and-public-health-preparedness/article/misinformation-in-wake-of-the-covid19-outbreak-fueling-shortage-and-misuse-of-lifesaving-drugs-in-pakistan/6048D98D3E44BAA3A3732343FE8C8A27), 3. Walkthrough Sanitization Gates for COVID-19: A Preventive Measure or Public Health Concern? (http://www.ajtmh.org/content/journals/10.4269/ajtmh.20-0533), 4. Drug repurposing for COVID-19: a potential threat of self-medication and controlling measures (https://www.ncbi.nlm.nih.gov/pmc/articles/PMC7448118/). Moreover, this manuscript requires extensive editing for English and syntax.

We look forward to receiving your revised manuscript.

Kind regards,

Tauqeer Hussain Mallhi, Ph.D

Academic Editor

PLOS ONE

Additional Editor Comments:

Thank you for submission to Plos One. This manuscript has been assessed by two experts and found it interesting. However, authors raised serious concerns in the readability, methodology and results. I invite you to consider these suggestions in point-by-point manner. Since COVID-19 is associated with massive infodemic resulting in various false beliefs, this manuscript is timely and well needed during this time, particularly in African regions. Myths and misleading beliefs during the pandemic has also raised serious concerns such as vaccine hesitancy, self-medication, inappropriate use of devices, drug shortages and price hikes. Various drafts have raised this issue but scientific evidence is currently lacking. I will suggest authors to consider following recently published articles in introduction and discussion section as most of the author`s claim are supported by them. 1. Threat of COVID-19 Vaccine Hesitancy in Pakistan: The Need for Measures to Neutralize Misleading Narratives (ajtmh.org/content/journals/10.4269/ajtmh.20-0654), 2. Misinformation in wake of the COVID-19 outbreak: Fueling shortage and misuse of lifesaving drugs in Pakistan (https://www.cambridge.org/core/journals/disaster-medicine-and-public-health-preparedness/article/misinformation-in-wake-of-the-covid19-outbreak-fueling-shortage-and-misuse-of-lifesaving-drugs-in-pakistan/6048D98D3E44BAA3A3732343FE8C8A27), 3. Walkthrough Sanitization Gates for COVID-19: A Preventive Measure or Public Health Concern? (http://www.ajtmh.org/content/journals/10.4269/ajtmh.20-0533), 4. Drug repurposing for COVID-19: a potential threat of self-medication and controlling measures (https://www.ncbi.nlm.nih.gov/pmc/articles/PMC7448118/). Moreover, this manuscript requires extensive editing for English and syntax.

Journal Requirements:

Reviewers' comments:

Reviewer's Responses to Questions

**Comments to the Author**

1. Is the manuscript technically sound, and do the data support the conclusions?

Reviewer #1: Partly

Reviewer #2: Yes

2. Has the statistical analysis been performed appropriately and rigorously? 

Reviewer #1: I Don't Know

Reviewer #2: Yes

3. Have the authors made all data underlying the findings in their manuscript fully available?

Reviewer #1: Yes

Reviewer #2: Yes

4. Is the manuscript presented in an intelligible fashion and written in standard English?

Reviewer #1: Yes

Reviewer #2: No

5. Review Comments to the Author

Reviewer #1: Review of study entitled, Myths, beliefs, and perceptions about COVID-19 in Ethiopia: The need to address information gaps Context of study

The study is timely and relevant for the country of study as well as the rest of Africa. However it needs strengthening in various areas.

1. I recommend professional editing of the entire manuscript. The entire manuscript needs improvement in language, as articles, wrong commas, wording, punctuations among others have been used

2. I am not sure the style of reporting figures and proportions in some parts of the manuscript is standard

3. Some of the wording in the tables need improvement

4. The three figures are blur and do not reflect nor represent the narrative. The authors also appear to assume that the figures will complement the narratives in some sections, however tables are equally required in those sections

5. How has COVID-19 been contained in Ethiopia such as government and community efforts with references from the relevant ministries, other stakeholders? The authors provide scanty report on government’s efforts in the prevention of COVID-19. Authors need to provide some reasonable detail on the Ethiopian government and relevant bodies intervention strategies and efforts in the fight against COVID-19 ,

6. Authors do not provide contextual information of the study regions such as level of urbanization/rural, educational levels, population estimates among others. A table on such details will be useful. To ensure that an online study is at least credible and useful such information is critical.

7. While it is no doubt an online study it is still important that authors disaggregate data to explain the situation between men and women, educated and illiterate to enable a better appreciation of the results.

8. Percentages are sometimes placed at the wrong places of sentences

Specific comments

• The use of etc. in some sections of the manuscript, please state all factors or use “among others”

• The repeated use of i.e. should be in full

• Line 133 correct “that slow down it”

Background characteristics of participants

• Disaggregate according to sex, age and educational level

• Why were women that few, which region had the most women and men and why?

Section on measurement and operationalization

Line 133 correct sentence “that slow down it”

Section on data base

Line 143, correct “Multi response analysis was performed for every perceptions.”

Table 2

• What does “People still use suffocated transportation” mean?

• Incomplete: “People do not often seek care for symptoms that looks like it”

• Note on “Kaiser Mayer Olkin’s measure of sampling adequacy”, if the name is an authority kindly reference appropriately.

Section on category of inhibitors

• Lines 191, authors should explain what these terms mean “religiosity”, “general reliance in unconfirmed traditional medicine”

• Lines 193-194, authors state “The second category of perceived inhibitor i.e false assurances was constructed from two beliefs: “we live far away from hot spot areas” and “ there are no cases reported in our locality”,

o What kinds of interviews were conducted, how were text analyzed?

o

o use proper citing with italics of quotation

o Authors should include the study questionnaires/guides as appendix

Section on prevalence of inhibitors

o Lines 201-203: Do not say anything

o What does the following mean? “Accordingly, the prevalence of specific beliefs that built myths ranged between (54.7%) and 140 (15.1%).”

o The same issues have been reported in the sections on Categories of inhibitors and Prevalence of inhibitors.

Section on Spatial distributions of the perceptions: variations by regions and townships

o This section does not mean anything without contextual information on the regions to help the reader to understand the context.

Township distribution and variation

o Authors should provide a table

o Line 297-299. Authors indicate seven but six were reported in this study, “seven commonest communication channels and 298 platforms were used for scoring access: Television, mobile data, social media, health workers, radio, and Wi-Fi.”

o “dot” in front of the following section title should be removed, “. Perception of threat and perceived facilitators, inhibitors, and information needs”

Discussion section

o Line 331-334, authors mention two studies but report on one as follows: “Two, in one of the previous studies conducted in Ethiopia, 179 (72.5%) %) of respondents knew that older ages and people with 333 underlying illnesses are high-risk groups, while only 15 (6.1%) knew that young adult people must engage 334 precautions just like any other segment of people (22).”

o Paragraph 336-360 seems more or less as a report than a discussion paragraph. Besides authors appear to use previously stated quotations that do not follow the standard reporting guidelines for quotations.

o Lines 403 to 411 does not appear to be a discussion paragraph as the study is not compared or contrasted with literature

o Lines 412-418 does not appear to be a discussion paragraph

Under limitation of study

o Authors state “To the best of our knowledge, this study is the first of its kind in reporting community perceptions and myths in Ethiopia.” Authors should correct it to include “first kind in community perceptions and myths on COVID-19”

Conclusion

o What do the following phrases mean “ownership of traditional medicines”; “”people with old ages“; a country like Ethiopia whose major portion is populated with this age segments"?

Reviewer #2: The manuscript needs typographical , grammatical errors and some sentence constructions corrections for more clarity , some of them are indicated below:-

- Line 75 &76 : Of 3, 961,425 closed cases,(10%) ended 76 up in deaths. (what are closed cases?)

- Line 98 & 99: WHO warns the investigation and control of “infodemics”, myths, and stigma, while fighting the pandemic 99 through appropriate risk communication and community engagement principles (the sentence lacks clarity)

- Line 101-104: Moreover, an up-to-date information needs regarding causes, means of protection, modes of transmissions, diagnostic symptoms, and treatment/isolation procedures are basic knowledge to withstand myths, an impression of invulnerability, and support preventive efforts (sentence lacks clarity)

- Line 174 : the phrase “suffocated transport means” , which also appears in several parts of the manuscript is an ambiguous phrase. Assuming that it is meant to describe “crowded unventilated transport means”, is it possible to change the phrase ?

- Line 222: …….and procedures to follow when felt symptomatic (correct "felt symptomatic")

- Line 364: Interestingly, the first two of the three factors were misperceived inhibitors i.e. why we labeled them as myths and false assurances (the use of the abbreviation “ i.e.” does not fit in this sentence, it is better to use the full phrase “that is “

- Line 407 : ……..of cases and zero death are found in the region till a moment of June 9, 2020 (instead of " till a moment of June 2020" use the phrase "as of June 9, 2020)

Additional comments and questions:

- There are 9 National regional states and two administrative states in Ethiopia. Table 1 (line 153) shows that majority of the respondents were from 4 regions and one of the administrative state (i.e Addis Ababa). This means that five out of the 9 regions and one administrative state is within the "others" which is only 6.5% ? Do you think this could be representative of all regions in the country, a country with diverse cultures and beliefs. Do you believe that the data allows you to interpret regional and township variations and thus affect your recommendation for communication and community engagement ? Are there any regions that were not included? If , yes, that data should be reflected

- Line 371 & 372: Please include a reference for this sentence

" Pieces of evidence indicate that myths or misperceptions can set back preventive and control efforts in times of crisis, and pandemics of HIV, Zikavirus, Yellow fever, Ebola, etc, unless traced and addressed ".

- Line 350 : One of the factors for enabling environmental conditions is : people do not have hand rub alcohol or sanitizers. Why was the question only focused on sanitizers and alcohol and why was the availability of water and soap not considered?

- As the authors have rightly indicated one major limitation of the study is the selection bias of educated participants who have access to internet , in addition to being a proxy indicator, But the authors ascertain that the findings are pertinent in that the respondents lived in the community that they represented. This argument is not convincing , since still the community that they represent might be limited to their own circle of educated people

6. PLOS authors have the option to publish the peer review history of their article (what does this mean?). If published, this will include your full peer review and any attached files.

Reviewer #1: No

Reviewer #2: No

---

## [Author Response · Author response to Decision Letter 0]

6 Nov 2020

Responses to reviewers: A rebuttal letter

Dear editor and reviewers (Profs./Drs), 

Many thanks for your review and comments to our manuscript on myths about COVID-19. We are so thankful for the insights, feedbacks, and articles you shared to strengthen our work. We hope that we have now addressed your concerns and questions in text the revised version. We provided point-by-point response to your comments, suggestion, and questions. We used track changes to mark where the changes are in the revised document. We also assure you that the manuscript style meets the PLOS ONE’s requirements. We provided additional information about the like questionnaire we adapted for use in this work. We also have presented ethics statement only in methods section. Please, follow our responses to the comments. The responses indicated lines where revisions were made as referred in to “manuscript with track changes”. 

Comments 

PONE-D-20-17606

Myths, beliefs, and perceptions about COVID-19 in Ethiopia: The need to address information gaps

PLOS ONE

Dear Dr. Kebede,

Thank you for submitting your manuscript to PLOS ONE. After careful consideration, we feel that it has merit but does not fully meet PLOS ONE’s publication criteria as it currently stands. Therefore, we invite you to submit a revised version of the manuscript that addresses the points raised during the review process.

Thank you for submission to Plos One. This manuscript has been assessed by two experts and found it interesting. However, authors raised serious concerns in the readability, methodology and results. I invite you to consider these suggestions in point-by-point manner. Since COVID-19 is associated with massive infodemic resulting in various false beliefs, this manuscript is timely and well needed during this time, particularly in African regions. Myths and misleading beliefs during the pandemic has also raised serious concerns such as vaccine hesitancy, self-medication, inappropriate use of devices, drug shortages and price hikes. Various drafts have raised this issue but scientific evidence is currently lacking. I will suggest authors to consider following recently published articles in introduction and discussion section as most of the author`s claim are supported by them. 1. Threat of COVID-19 Vaccine Hesitancy in Pakistan: The Need for Measures to Neutralize Misleading Narratives (ajtmh.org/content/journals/10.4269/ajtmh.20-0654), 2. Misinformation in wake of the COVID-19 outbreak: Fueling shortage and misuse of lifesaving drugs in Pakistan (https://www.cambridge.org/core/journals/disaster-medicine-and-public-health-preparedness/article/misinformation-in-wake-of-the-covid19-outbreak-fueling-shortage-and-misuse-of-lifesaving-drugs-in-pakistan/6048D98D3E44BAA3A3732343FE8C8A27), 3. Walkthrough Sanitization Gates for COVID-19: A Preventive Measure or Public Health Concern? (http://www.ajtmh.org/content/journals/10.4269/ajtmh.20-0533), 4. Drug repurposing for COVID-19: a potential threat of self-medication and controlling measures (https://www.ncbi.nlm.nih.gov/pmc/articles/PMC7448118/). Moreover, this manuscript requires extensive editing for English and syntax.

We look forward to receiving your revised manuscript.

Kind regards,

Tauqeer Hussain Mallhi, Ph.D

Academic Editor

PLOS ONE

Responses

Comment 1.

Additional Editor Comments:

Thank you for submission to Plos One. This manuscript has been assessed by two experts and found it interesting. However, authors raised serious concerns in the readability, methodology and results. I invite you to consider these suggestions in point-by-point manner. Since COVID-19 is associated with massive infodemic resulting in various false beliefs, this manuscript is timely and well needed during this time, particularly in African regions. Myths and misleading beliefs during the pandemic has also raised serious concerns such as vaccine hesitancy, self-medication, inappropriate use of devices, drug shortages and price hikes. Various drafts have raised this issue but scientific evidence is currently lacking. I will suggest authors to consider following recently published articles in introduction and discussion section as most of the author`s claim are supported by them. 1. Threat of COVID-19 Vaccine Hesitancy in Pakistan: The Need for Measures to Neutralize Misleading Narratives (ajtmh.org/content/journals/10.4269/ajtmh.20-0654), 2. Misinformation in wake of the COVID-19 outbreak: Fueling shortage and misuse of lifesaving drugs in Pakistan (https://www.cambridge.org/core/journals/disaster-medicine-and-public-health-preparedness/article/misinformation-in-wake-of-the-covid19-outbreak-fueling-shortage-and-misuse-of-lifesaving-drugs-in-pakistan/6048D98D3E44BAA3A3732343FE8C8A27), 3. Walkthrough Sanitization Gates for COVID-19: A Preventive Measure or Public Health Concern? (http://www.ajtmh.org/content/journals/10.4269/ajtmh.20-0533), 4. Drug repurposing for COVID-19: a potential threat of self-medication and controlling measures (https://www.ncbi.nlm.nih.gov/pmc/articles/PMC7448118/). Moreover, this manuscript requires extensive editing for English and syntax.

Response 1

Dear Editor, thank you for evaluating this work as timely and is important. We are indebted to you for sharing these pertinent studies. We have used them all in the discussion sections of the revised version (please check reference # 27, 32, 33, and 35). We also have improved the writing in English using grammarly and R Pubsure software online. 

 Comment 2

Journal Requirements:

 Response 2

We are thankful for sharing the Journal’s requirements and styles. We have checked, the following: 

1. We checked the journal’s requirements in the revised version, including formatting issues. 

2. We have provided additional information about the study. The questionnaire we used is annexed as supplementary file in revised submission. 

3. We have provided ethics statement only in methods section of the revised version of the manuscript. 

Reviewers' comments 3:

Reviewer's Responses to Questions

Comments to the Author

1. Is the manuscript technically sound, and do the data support the conclusions?

Reviewer #1: Partly

Reviewer #2: Yes

2. Has the statistical analysis been performed appropriately and rigorously?

Reviewer #1: I Don't Know

Reviewer #2: Yes

3. Have the authors made all data underlying the findings in their manuscript fully available?

Reviewer #1: Yes

Reviewer #2: Yes

4. Is the manuscript presented in an intelligible fashion and written in standard English?

Reviewer #1: Yes

Reviewer #2: No

Response 3

Thank you for your evaluation. We believe that the conclusion based entirely based on the study. In fact, we have improved specific comments provided regarding conclusion. We also believe we have copyedited the manuscript for English using online accessed software like grammarly and R Pubsure. 

5. Review Comments to the Author

Reviewer 1 general comments 

Comment 4

Reviewer #1: Review of study entitled, Myths, beliefs, and perceptions about COVID-19 in Ethiopia: The need to address information gaps Context of study

The study is timely and relevant for the country of study as well as the rest of Africa. However it needs strengthening in various areas.

1. I recommend professional editing of the entire manuscript. The entire manuscript needs improvement in language, as articles, wrong commas, wording, punctuations among others have been used

Response 4: We are so grateful to Prof/Dr. reviewer 1 for his/her priceless comments that strengthened the manuscript. We also thank for your view of this study as timely and worthy of investigation even for the rest of Africa. Regarding improvement of its English writing, we believe we have now improved the document using online software. 

Comment 5

2. I am not sure the style of reporting figures and proportions in some parts of the manuscript is standard

Response 5: Thank you for your concern regarding reporting figures and proportions. The figures were drawn for latent variables constructed from summations of specific variables in that particular factor. Then, for easy comparison all variables were converted to 100%. So, line graphs reported scores of that particular factor out of 100. Therefore, we can confirm you that the figures and proportions used are accurate and carefully analyzed. 

Comment 6

3. Some of the wording in the tables needs improvement

Response 6: Thank you for your comment. Although this comment is general and non-specific, we have tried to have improved some wording where we felt was appropriate in the table. 

Comment 7:

4. The three figures are blur and do not reflect nor represent the narrative. The authors also appear to assume that the figures will complement the narratives in some sections, however tables are equally required in those sections

Response 7: Regarding blurring effect of the figure, I think the quality goes to the SPSS software we are using. Nonetheless, we can confirm you that the figures were uploaded into the PACE (the journal’s requirement for figure) and has successfully passed it. So we can warrant you that the journal itself don’t accept the figures that don’t fulfill the journals’ minimum requirement. Regarding your comment that tables are required in addition to the figures, we are thankful. However, we argue that using tables in addition to figures will increase redundancy while the added value is minimal. We believed that the figures carried significant information about distribution of myths and other perceptions across regions and towns. In fact, table 6 carried descriptive statistics of the main concepts that latter appeared in figures about regional distributions. Therefore, we politely ask you accept our response. 

Comment 8

5. How has COVID-19 been contained in Ethiopia such as government and community efforts with references from the relevant ministries, other stakeholders? The authors provide scanty report on government’s efforts in the prevention of COVID-19. Authors need to provide some reasonable detail on the Ethiopian government and relevant bodies intervention strategies and efforts in the fight against COVID-19,

Response 8: We are so thankful for your comment. We have inserted some lines in the introduction section referring to what government in Ethiopia has been doing to contain the virus. For example, we have inserted (by current lines 117-122 on manuscript with track changes)the following statements in the introduction section, “At the moment of the study, the government has declared state emergency in support of the precautionary measures, and has taken public measures such as: closure of schools including Universities; installation of locally available technologies at public service outlets, including hand washing basins; limiting number of passengers in public transports, among others. Moreover, the ministry of health engaged on public awareness creation, risk communication, and community engagement tasks supported by advisory councils established from recognized Universities.” 

Comment 9: 

6. Authors do not provide contextual information of the study regions such as level of urbanization/rural, educational levels, population estimates among others. A table on such details will be useful. To ensure that an online study is at least credible and useful such information is critical.

Response 9: Thank you very much this comment. We have now included in “study setting sub-section of the methods section” contextual information about the regions, overall literacy, and population estimates. But we used only text to include the information, not actually table. Refer, to the information in specified sub-section, please. The following statement added by lines 130-137, “At the time of the study, administratively, Ethiopia is divided into nine regional states and two federal cities. The regions have zonal divisions and district sub-divisions, with respective regional capitals and zonal/district twons. Internet services are rarely accessible at district level but mobile data. Ethiopia 2020 population is estimated at 114,963,588 people at mid-year according to UN data. 21.3% of the population is urban (24,463,423 people in 2020). The median age in Ethiopia is 19.5 years. Ethiopian 2020 average literacy rate is 49.1% (lower among adults: male, 57.2; female, 41.1%, and higher among youths: male, 71.1%; female,67.8%).

Comment 10

7. While it is no doubt an online study it is still important that authors disaggregate data to explain the situation between men and women, educated and illiterate to enable a better appreciation of the results.

Response 10: We are thankful for your comment. Detail report about background of participants was reported in separate article, which we believe it will shortly be published soon. As indicated in table 1, the women are disproportionately low in this study, only 10%. Plus, given an online study no illiterate people involved in this study. We emphasized in picking existing myths and perceptions. Please, our apologies we have presented more details in separate article. We would love to avoid redundancy. 

Comment 11

8. Percentages are sometimes placed at the wrong places of sentences

Response 11: We have checked percentages that were out at the wrong places of sentences, and corrected which ever we noticed. 

Specific comments

Comment 12: 

• The use of etc. in some sections of the manuscript, please state all factors or use “among others”

• The repeated use of i.e. should be in full

• Line 133 correct “that slow down it”

Response 12: Thank you for your comments. We corrected them all. 

• We have removed the use of “etc”. Instead, we used the list of all factors or “among others” in the revised version. 

• The used of “i.e” changed to full text “that is/are/was”

• Line 133 (currently by line 157), we corrected “slow down it” into “….slow down the spread of COVID-19”. 

Comment 13

Background characteristics of participants

• Disaggregate according to sex, age and educational level

• Why were women that few, which region had the most women and men and why?

Response 13: 

o We have recently provided the response to this comment; refer to response #10 above. With limitation of online as is, we would like to report the myths and beliefs belong to community. All participants are educated, and we didn’t intend to look variations by education or age. 

o Perhaps, females participates were low because 1) literacy generally is lower among females, and 2) most respondents for this study were from higher academic institutions were females’ involvement generally is very low compared to males. 

 Comment 14 

Section on measurement and operationalization

Line 133 correct sentence “that slow down it”

Response 14: corrected, as responded already by response # 13. (check current line 157)

Comment 15

Section on data base

Line 143, correct “Multi response analysis was performed for every perceptions.”

Table 2

• What does “People still use suffocated transportation” mean?

• Incomplete: “People do not often seek care for symptoms that looks like it”

• Note on “Kaiser Mayer Olkin’s measure of sampling adequacy”, if the name is an authority kindly reference appropriately.

Response 15: Thank you for the comments given to table 2, we revised the manuscript accordingly, 

• Line 143 (current line 168) we corrected the statement by inseting article “a” and removed “s” from perceptions.

In Table 2

• We have changed “people still use suffocated transport” into “people still use crowded transport means”. 

• We completed the statement “People do not often seek care for symptoms that looks like it” by adding “ COVID-19 instead of ‘it’. 

• Regarding “Kaiser Mayer Olkin’s measure of sampling adequacy” we added a sentence about it in methods section, ‘measurement and operationalization’ sub-section. We also included citation # 24. We did this because it is uncommon to include citation in results. We included a statement that KMO>50% indicate the sample is adequate for running EFA. Under table 2, we now used abbreviation, KMO.

Comment 16:

Section on category of inhibitors

• Lines 191, authors should explain what these terms mean “religiosity”, “general reliance in unconfirmed traditional medicine”

• Lines 193-194, authors state “The second category of perceived inhibitor i.e false assurances was constructed from two beliefs: “we live far away from hot spot areas” and “ there are no cases reported in our locality”,

o What kinds of interviews were conducted, how were text analyzed?

o use proper citing with italics of quotation

o Authors should include the study questionnaires/guides as appendix

Response 16: Thank you for your careful review on category of inhibitors, the specific comments were explained and corrected, accordingly.

• Line 191 (the current line 218) the term “religiosity” is referred to the extent to which people feel themselves as religious enough to be able to effectively manage challenges in their faith. We latter used “perceived religiosity” so we stick to “perceived religiosity” throughout the revision. The variable included in table 3 can provide self-explanation to the term we used as “religiosity”, that is, “we are religious enough to control COVID-19”. We have explained in bracket by line 218 that ‘perceived religiosity” meat that “perceiving oneself as effective religious man/woman in controlling challenges). We also have changed “general reliance in unconfirmed traditional medicine” to “people’s perceived confidence that they owned effective traditional medicines that are, however, not clinically confirmed”

• Line 193-194 (current line 222-226). Similar to the above response, we have improved the statements as such, “The second category of perceived inhibitors was still local sayings that were often related to people’s false assurances that they were protected from COVID-19 (unlike myths, the second category of beliefs may not need scientific approval or disapproval). The category consisted of two main beliefs: “we live far away from COVID-19 rampant areas” and “ there are no locally reported COVID-19 cases so far ”

• Regarding interviews conducted, we have plainly put that this was an online cross-sectional survey conducted using questionnaire. Sometimes, we left open spaces, to type their responses. Any responses about perceptions of facilitators or inhibitor was considered as variable and counted.

• Quotations reported in italics in the revised version. 

• We have included questionnaire as supplementary files

Comment 17

Section on prevalence of inhibitors

o Lines 201-203: Do not say anything

o What does the following mean? “Accordingly, the prevalence of specific beliefs that built myths ranged between (54.7%) and 140 (15.1%).”

o The same issues have been reported in the sections on Categories of inhibitors and Prevalence of inhibitors.

Response 17: Thank you for your comments on this section

o Lines 201-203 (current lines234-238) were improved to convey clear idea as such: “Myths and false assurances were the most prevalent perceived inhibitors of the spread of COVID-19 compared the perception that engagement in precautionary measures protect from exposure to and spread of the virus. Specifically, perceived religiosity, perceived effectiveness of selected foods, and perceived protectiveness of hot weather were the commonest myths, accounting 508 (54.7%), 455 (49.0%), and 242 (26.0%), respectively. Beliefs that there were no locally reported cases of COVID-19 and the specific localities where respondents are currently living are far away from corona virus rampant areas contributed to 343 (36.9%) and 274 (29.5%) prevalence of false assurances”

Comment 18:

Section on Spatial distributions of the perceptions: variations by regions and townships

o This section does not mean anything without contextual information on the regions to help the reader to understand the context.

Response 18: We have now provided contextual information about regions to help understand the context in study setting sub-section in methods. We also noted about this in our response # 9. 

Comment 19: 

Township distribution and variation

o Authors should provide a table

o Line 297-299. Authors indicate seven but six were reported in this study, “seven commonest communication channels and 298 platforms were used for scoring access: Television, mobile data, social media, health workers, radio, and Wi-Fi.”

o “dot” in front of the following section title should be removed, “. Perception of threat and perceived facilitators, inhibitors, and information needs”

Response 19: We are thankful for your critical review and comments.

o Regarding table we also provided response to this comment in our response #7. The added value of inserting table is minimal. We have presented the required data through the line graphs. Adding table mostly causes redundancy of data presentation. 

o Line 297-299 (current lines 247-248) we added, “broad band internet service”. Now, the list is seven. 

o By current line 356, We removed the “dot” in front of the section title ““. Perception of threat and perceived facilitators, inhibitors, and information needs”

 Comment 20:

Discussion section

o Line 331-334, authors mention two studies but report on one as follows: “Two, in one of the previous studies conducted in Ethiopia, 179 (72.5%) %) of respondents knew that older ages and people with 333 underlying illnesses are high-risk groups, while only 15 (6.1%) knew that young adult people must engage 334 precautions just like any other segment of people (22).”

o Paragraph 336-360 seems more or less as a report than a discussion paragraph. Besides authors appear to use previously stated quotations that do not follow the standard reporting guidelines for quotations.

o Lines 403 to 411 does not appear to be a discussion paragraph as the study is not compared or contrasted with literature

o Lines 412-418 does not appear to be a discussion paragraph

Response 20: Thank you for your comment on the discussion section. The comments were addressed in the revised version. 

o Line 331-334 (current line 387-389). In a paragraph stating the said concepts, we did not say two studies. We did say a couple reasons opposed to two studies. Please check. Reference 21 and 22 were cited for reason number 1 and 2. 

o Line 336-360 (current line 392-421): Regarding discussions in those lines, we provided conceptual discussion. For example, a factor with “lack of enabling environment” would conceptual mean “the need for much resources” for COVID-19, and “behavioral non-adherence” would mean “social ignorance and lacks of commitment” . We have supplemented some theoretical explanation as what does these mean in terms of behavioral and communication theories and WHO’s preparedness and response with regard to community engagement and risk communication. Statements that have sense of quotations were removed from this paragraph. In fact, we did not use any quotations. We used variables that illustrated the themes in which they belonged. 

o Line 403-411 (current line 474-484). Obviously literatures are limited on this study. We have now compared the findings against available resources. We have used reference # 28, 29 to explain what these findings mean. We also have discussed these in terms of the geographical presence and interactions between the regions. 

o Line 412-418 (current line 485-497): Again, we emphasized on conceptual meaning of it. In fact, we also have used reference #29 in order to support our argument. At the same time, we added one more study with false assurance cited as # 35. 

Comment 21

Under limitation of study

o Authors state “To the best of our knowledge, this study is the first of its kind in reporting community perceptions and myths in Ethiopia.” Authors should correct it to include “first kind in community perceptions and myths on COVID-19”

Response 21: Thank you very much, we accepted the comment and changed the statement accordingly. 

o The revised statement reads, “To the best of our knowledge, this study is the first kind of community perceptions and myths on COVID-19 in Ethiopia” refer by line 519-20. 

Comment 22: 

Conclusion

o What do the following phrases mean “ownership of traditional medicines”; “”people with old ages“; a country like Ethiopia whose major portion is populated with this age segments"?

Response 22: Thank you for your careful review of this manuscript. We addressed the comment in the revised version. 

o We have revised our conclusion regarding statements/phrases about “ownership of medicine, people with old ages, and Ethiopian population”. We deleted “ownership”, used “elderly people”, and “ a country with high percentage of young population, like Ethiopia” instead of the previous phrases. 

Reviewer 2 comments 

Comment 23

Reviewer #2: The manuscript needs typographical , grammatical errors and some sentence constructions corrections for more clarity , some of them are indicated below:-

- Line 75 &76 : Of 3, 961,425 closed cases,(10%) ended 76 up in deaths. (what are closed cases?)

- Line 98 & 99: WHO warns the investigation and control of “infodemics”, myths, and stigma, while fighting the pandemic 99 through appropriate risk communication and community engagement principles (the sentence lacks clarity)

- Line 101-104: Moreover, an up-to-date information needs regarding causes, means of protection, modes of transmissions, diagnostic symptoms, and treatment/isolation procedures are basic knowledge to withstand myths, an impression of invulnerability, and support preventive efforts (sentence lacks clarity)

- Line 174 : the phrase “suffocated transport means” , which also appears in several parts of the manuscript is an ambiguous phrase. Assuming that it is meant to describe “crowded unventilated transport means”, is it possible to change the phrase ?

- Line 222: …….and procedures to follow when felt symptomatic (correct "felt symptomatic")

- Line 364: Interestingly, the first two of the three factors were misperceived inhibitors i.e. why we labeled them as myths and false assurances (the use of the abbreviation “ i.e.” does not fit in this sentence, it is better to use the full phrase “that is “

- Line 407 : ……..of cases and zero death are found in the region till a moment of June 9, 2020 (instead of " till a moment of June 2020" use the phrase "as of June 9, 2020)

Response 23: Thank you very much, indeed for your careful reviewing of our manuscript. We have addressed all your comments in the revised version point-by-point as follows:

-Line 75 and 76 (current line 80-81): In the statement, “Of 3, 961,425 closed cases,(10%) ended 76 up in deaths”. Closed cases mean cases of COVID-19 that resulted in discharge either because of cure or deaths. We used “closed cases” as referred by worldometer. We believe using “closed cases” as it is will be good give that worldometer dashboard for COVID-19 uses the same term. 

-- Line 98 & 99 (current line 105-108): The statement. “WHO warns the investigation and control of “infodemics”, myths, and stigma, while fighting the pandemic 99 through appropriate risk communication and community engagement principles” was changed to, “At this moment of the pandemic, WHO recommends the risk communication and community engagement efforts to investigation and control “infodemics”, myths, beliefs, and stigma so that the spread of the spread of corona virus would be appropriatel combated” in the revised version. 

- Line 101-104 (current line 110-113): The statement, “ Moreover, an up-to-date information needs regarding causes, means of protection, modes of transmissions, diagnostic symptoms, and treatment/isolation procedures are basic knowledge to withstand myths, an impression of invulnerability, and support preventive efforts” was changed to “Moreover, an up-to-date information regarding causes, means of protection, modes of transmissions, diagnostic symptoms, and treatment/isolation procedures are basic to withstand myths, beliefs, perceptions, and support preventive efforts”. 

-Line 174 (current line 189): the phrase, “suffocated transport means” was changed to “crowded unventilated transport means”, as you suggested, and across the manuscript. 

- Line 222 (current line 263): as suggested we corrected "felt symptomatic" to “symptomatic” we removed “felt”

- Line 364 (current line xxx): we used full phrase for “i.e.” in a statement “Interestingly, the first two of the three factors were misperceived inhibitors i.e. why we labeled them as myths and false assurances”

- Line 407 (current line 477) : the comment to change “till a moment of June 2020” into “as of June 9, 2020” in a statement, “……..of cases and zero death are found in the region till a moment of June 9, 2020 (instead of " till a moment of June 2020" needs some clarification. We didn’t want to imply “as of June 9, 2020”. Instead, we wanted “until June 9, 2020”. Thus, we modified the phrase in that statement into, “……..until June 9, 2020”

Comment 24: Additional comments and questions:

- There are 9 National regional states and two administrative states in Ethiopia. Table 1 (line 153) shows that majority of the respondents were from 4 regions and one of the administrative state (i.e Addis Ababa). This means that five out of the 9 regions and one administrative state is within the "others" which is only 6.5% ? Do you think this could be representative of all regions in the country, a country with diverse cultures and beliefs. Do you believe that the data allows you to interpret regional and township variations and thus affect your recommendation for communication and community engagement ? Are there any regions that were not included? If , yes, that data should be reflected

- Line 371 & 372: Please include a reference for this sentence

" Pieces of evidence indicate that myths or misperceptions can set back preventive and control efforts in times of crisis, and pandemics of HIV, Zikavirus, Yellow fever, Ebola, etc, unless traced and addressed ".

- Line 350 : One of the factors for enabling environmental conditions is : people do not have hand rub alcohol or sanitizers. Why was the question only focused on sanitizers and alcohol and why was the availability of water and soap not considered?

- As the authors have rightly indicated one major limitation of the study is the selection bias of educated participants who have access to internet , in addition to being a proxy indicator, But the authors ascertain that the findings are pertinent in that the respondents lived in the community that they represented. This argument is not convincing , since still the community that they represent might be limited to their own circle of educated people

Response 24: Thank you for your key comments and questions. We have clarified them. We also have modified the manuscript based on the comments. 

-Regarding regions and cities included in the study, let us explain it. The study has not excluded any region or city. We have now provided information about regions and towns in the study setting sub-sections. During analysis, we merged regional or federal cities as big towns. Many regions like Somali, Benishangul, Gambela, Afar were merged as “other regions”. Honestly speaking, we didn’t intend to merge any town or region at the moment we started the online data collection. Because of the urgent need of the study to inform risk communication and community engagement, we closed the data collection within 2 weeks, after getting some sample of 929 respondents across the country. As we think now, the problem was that we closed the data collection early. Unfortunately, during analysis we observed that the involvement of the above merged regions and cities were low. So, we were forced to merge some of them. We appreciate the critical view provided to us regarding adequacy of representations. We include a statement, “Although the study was nationwide, participation from some regions were limited compared to others. Perhaps, extended data collection period would have increased their involvement and representations “ in the limitation section of the study. However, we still believe the analysis reveals pertinent finding with exception of the merging of some. Regarding towns, we considered sizes and communication opportunities. So, we don’t think that would be a big problem. Regarding, recommendation we have shown two points: 1) to check regional variations in addressing myths and false assurances, 2) further investigation of beliefs and myths through additional assessments. The limitation would be embraced here. 

-- Line 371 & 372 (current line 440-444): Four references for this sentence, " Pieces of evidence indicate that myths or misperceptions can set back preventive and control efforts in times of crisis, and pandemics of HIV, Zikavirus, Yellow fever, Ebola, etc, unless traced and addressed ". The new reference # 28-31 were about this. 

-- Line 350 (current line 408): Regarding inclusion of water and soap in this study, we would say two points. 1) there was open space, where respondents can add more. Unfortunately, soap and water were not touched in their open response. 2) we did not put this as major list of factor, (we did mistake in missing that). Perhaps, we have previously published on knowledge and practice about COVID-19 as used in reference number 25. In that study, the dominantly practiced behavior was hand washing with soap and water. That could have affected us unknowingly to make water/soap on list of options. Moreover, that the moment of this study, water/soap was found in every corner of towns and it was less likely to bear in mind. Still, we don’t imply that was a good idea to include soap/water in this study to know how that was prevalent lack as COVID-19 preventive resource. Thank you once again. 

-Thank you once more for critical ideas you always raised toward strengthening this manuscript. Regarding the comment you provided on limiting community in the circle of their educated people, we have completely saluted. We have now removed that idea from the limitation of the study. It is true mentioning “communities are represented by their educated people” is conflicting with previously mentioned idea, “only educated people are more likely to access internet”. 

6. PLOS authors have the option to publish the peer review history of their article (what does this mean?). If published, this will include your full peer review and any attached files.

Do you want your identity to be public for this peer review? For information about this choice, including consent withdrawal, please see our Privacy Policy.

Reviewer #1: No

Reviewer #2: No

 Comment 25: 

While revising your submission, please upload your figure files to the Preflight Analysis and Conversion Engine (PACE) digital diagnostic tool, https://pacev2.apexcovantage.com/. PACE helps ensure that figures meet PLOS requirements. To use PACE, you must first register as a user. Registration is free. Then, login and navigate to the UPLOAD tab, where you will find detailed instructions on how to use the tool. If you encounter any issues or have any questions when using PACE, please email PLOS at figures@plos.org. Please note that Supporting Information files do not need this step

Response 25: The figures we uploaded even at the initial submission was passed the PACE criteria. They meet the PLOS’s requirement.

---

## [Decision Letter · Decision Letter 1]

16 Nov 2020

Myths, beliefs, and perceptions about the spread of COVID-19 in Ethiopia:  A need to address information and community engagement gaps

PONE-D-20-17606R1

Dear Dr. Kebede,

We’re pleased to inform you that your manuscript has been judged scientifically suitable for publication and will be formally accepted for publication once it meets all outstanding technical requirements.

Kind regards,

Tauqeer Hussain Mallhi, Ph.D

Academic Editor

PLOS ONE

Reviewers' comments:

Reviewer's Responses to Questions

**Comments to the Author**

1. If the authors have adequately addressed your comments raised in a previous round of review and you feel that this manuscript is now acceptable for publication, you may indicate that here to bypass the “Comments to the Author” section, enter your conflict of interest statement in the “Confidential to Editor” section, and submit your "Accept" recommendation.

Reviewer #1: All comments have been addressed

Reviewer #2: All comments have been addressed

2. Is the manuscript technically sound, and do the data support the conclusions?

Reviewer #1: Yes

Reviewer #2: Yes

3. Has the statistical analysis been performed appropriately and rigorously? 

Reviewer #1: I Don't Know

Reviewer #2: Yes

4. Have the authors made all data underlying the findings in their manuscript fully available?

Reviewer #1: Yes

Reviewer #2: Yes

5. Is the manuscript presented in an intelligible fashion and written in standard English?

Reviewer #1: Yes

Reviewer #2: Yes

6. Review Comments to the Author

Reviewer #1: The authors have responded to all the issues raised adequately and for that matter I have no further comments,

Reviewer #2: (No Response)

7. PLOS authors have the option to publish the peer review history of their article (what does this mean?). If published, this will include your full peer review and any attached files.

Reviewer #1: No

Reviewer #2: No

---

## [Editor Report · Acceptance letter]

18 Nov 2020

PONE-D-20-17606R1 

Myths, beliefs, and perceptions about COVID-19 in Ethiopia:  A need to address information gaps and enable combating efforts 

Dear Dr. Kebede:

I'm pleased to inform you that your manuscript has been deemed suitable for publication in PLOS ONE. Congratulations! Your manuscript is now with our production department. 

Kind regards, 

on behalf of

Dr. Tauqeer Hussain Mallhi 

Academic Editor

PLOS ONE